# Carbon neutral hydrogen storage and release cycles based on dual-functional roles of formamides

Duo Wei [1,2], Xinzhe Shi [1,2], Henrik Junge [2] ✉, Chunyu Du [1] ✉ & Matthias Beller [2] ✉

The development of alternative clean energy carriers is a key challenge for our society. Carbon-based hydrogen storage materials are well-suited to undergo reversible (de)hydrogenation reactions and the development of catalysts for the individual process steps is crucial. In the current state, noble metal-based catalysts still dominate this field. Here, a system for partially reversible and carbon-neutral hydrogen storage and release is reported. It is based on the dual-functional roles of formamides and uses a small molecule Fe-pincer complex as the catalyst, showing good stability and reusability with high productivity. Starting from formamides, quantitative production of CO-free hydrogen is achieved at high selectivity ( > 99.9%). This system works at modest temperatures of 90 °C, which can be easily supplied by the waste heat from e.g., proton-exchange membrane fuel cells. Employing such system, we achieve >70% $H_2$ evolution efficiency and >99% $H_2$ selectivity in 10 charge-discharge cycles, avoiding undesired carbon emission between cycles.

In the coming decades, society will experience a massive increase in the demand for renewable energy, specifically wind and solar, and to reduce carbon emissions caused by the combustion of fossil fuels[1]. To provide a reliable energy supply and more specifically to meet peak energy demands in densely populated regions as well as to avoid high electricity cost spikes, efficient ways storing fluctuating solar and wind power in both short and long terms are required. Besides classic mechanical approaches to store electric energy, with hydroelectric dams being the most famous ones[2], its conversion to chemical energy is discussed to be a feasible approach[3]. Here, hydrogen which can be easily produced by water electrolysis stands out as a means of an established commercial technology[4,5]. However, handling large quantities of hydrogen is troublesome, since the compressed gaseous and liquid $H_2$ requires vessels that can withstand high pressures (700 bar) and/or low temperatures (−253 °C) to achieve considerable hydrogen storage capacity. Such methods lead to high energy costs and require specific materials and equipment despite their good $H_2$ recovery[6]. Alternatively, chemical hydrogen storage-release methods convert $H_2$ to stable carrier molecules that can be stored and transported at ambient conditions and deliver afterward the stored $H_2$ on demand via dehydrogenation[7,8]. Such technologies could bridge the production of green $H_2$ from renewable electricity and its utilization in proton-exchange membrane (PEM) fuel cells to regenerate the stored renewable electricity for terminal energy consumption (Fig. 1a).

Besides the gaseous $H_2$ carriers e.g., ammonia[9] and methane[10], liquid organic hydrogen carriers (LOHC) offer high reversibility and superior kinetics in (de)hydrogenation, suitable for long distance transport and onboard applications[11,12]. As well-known examples of C1 compounds[13], methane[10], methanol[14], and formic acid (FA)[14,15] have been widely studied concerning hydrogen storage. Compared to ammonia ($\Delta G^0$ = +33.3 kJ $mol^{-1}$), methane ($\Delta G^0$ = + 113.6), and methanol ($\Delta G^0$ = +8.9 kJ $mol^{-1}$), formic acid (FA) ($\Delta G^0$ = −32.9 kJ $mol^{-1}$) is more thermodynamically favored in $H_2$ production process. Therefore, chemical $H_2$ storage-release cycles applying the $H_2$/$CO_2$-FA system have been well-developed in the past decades by using the greenhouse gas $CO_2$[16,17]. In addition, an intrinsically similar approach including bicarbonate and formate salts has also been investigated in reversible (de) hydrogenation processes ($\Delta G^0$ = ±0.7 kJ $mol^{-1}$)[18–20]. Surprisingly,

[1]School of Chemistry and Chemical Engineering, Harbin Institute of Technology, Harbin 150001, P. R. China. [2]Leibniz-Institut für Katalyse e.V, 18059 Rostock, Germany. ✉e-mail: henrik.junge@catalysis.de; cydu@hit.edu.cn; matthias.beller@catalysis.de

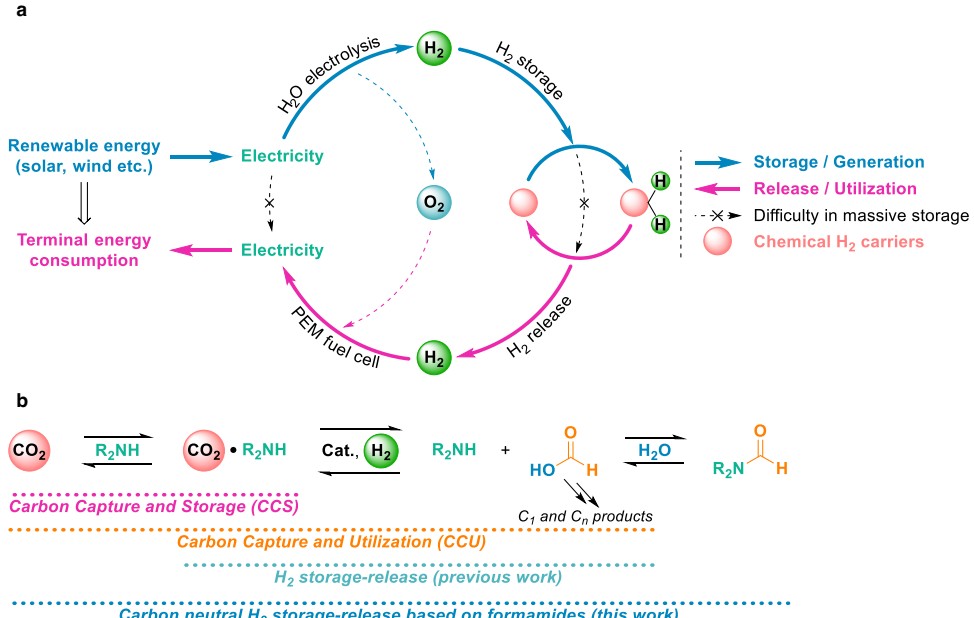

**Fig. 1 | Projected sustainable energy utilization based on renewable electricity storage and regeneration bridged by chemical hydrogen storage-release.**
**a** Renewable electricity can be converted to chemical fuel $H_2$ via water electrolysis. The resulting $H_2$ is easily transformed into stable chemical $H_2$ carriers for short- and long-term storage and transportation. The stored $H_2$ can be released on request to regenerate the renewable electricity in proton-exchange membrane (PEM) fuel cells. **b** Schematic illustration of amine-based carbon capture and storage (CCS), carbon capture and utilization (CCU), previously reported $H_2$ storage-release, and the strategy of carbon neutral $H_2$ storage-release based on dual-functional roles of formamides described in this work.

formamides as another class of easily and commercially available C1 compounds derived from $CO_2$ reduction in the presence of amines have been rarely studied directly in $H_2$ storage-release cycles[21–23].

It's worth noting that as $CO_2$ capturing reagents amines are frequently used in carbon capture and storage (CCS) processes[24], and further utilization of the $CO_2$-amine adducts (captured $CO_2$) in subsequent hydrogenation allows to produce renewable fuels and chemicals, so called carbon capture and utilization (CCU)[25]. As one of the most prominent examples of CCU, the "George Olah Methanol Plant" in Iceland is based on local renewable energy and $CO_2$[26]. Its total electrical energy demand and the overall efficiency reach 9.5 MWh/t methanol and 60%. Notably, such CCU processes also provide feasible approaches for sustainable chemical $H_2$ storage-release applications based on interconversion of $CO_2$ and C1 compounds (e.g., FA; Fig. 1b)[8]. For example, recently our group developed a reversible $H_2$ storage-release method based on amino acid lysine promoted $CO_2$ capture and its reversible hydrogenation to FA[17]. On the other hand, FA in the presence of amines could be easily dehydrated to formamides[27] which combine the carbon capturing reagent amines with the $H_2$ storage material FA. Therefore, the direct use of formamides as $H_2$ carriers is practically desired due to their dual-functional roles: the structurally incorporated FA is responsible for $H_2$ storage-release, and the built-in amines provide a carbon capture and utilization (CCU) strategy leading to an ultimate carbon neutral $H_2$ storage-release system (Fig. 1b). Compared to the hydrogen contents of FA (4.34 wt%), the ones of various formamides (1.50–3.17 wt%) are slightly lower but still higher than that of common formate salts (1.02–2.85 wt%, Fig. 2a). Bearing in mind that equivalent $CO_2$ is emitted together with $H_2$ in FA dehydrogenation process which generally requires a post carbon capturing process to reduce carbon emissions[14], in addition, $H_2$ storage using formate produces bicarbonate salts which could be decomposed to $CO_2$ as frequently reported[20]. Besides, another $H_2$ storage technology using $H_2$ storage alloys, e.g. magnesium hydrides[28], generally represent hydrogen contents of 1–6 wt%. However, their inferior (de)hydrogenation kinetics, life cycle, and harsh operation conditions (up to 500 °C) make them currently not suitable for most of the applications[28,29].

So far, expensive noble metal-based catalysts still dominate the area of $H_2$ storage and release. Therefore, the search of suitable non-noble metal catalysts and their efficient utilization in reversible $H_2$ storage-release cycles are particularly important. As a class of versatile catalysts, iron-based pincer complexes[14,30–42] have been studied respectively in hydrogenation[43–48] and dehydrogenation[49–51], attracting many interests for potentially reversible $H_2$ storage-release applications[37,38,52–54]. Owing to the metal-ligand cooperation effect, tridentate pincer complexes with a nitrogen donor (N-H) offer effective and stable catalysis in both hydrogenation and dehydrogenation steps[55–57]. As representatives, iron pincer complexes are used in $CO_2$ hydrogenation to produce FA (or its formate salts)[58–62], formamides[27], and methanol[63], as well as the $H_2$ production from FA[64–72] and methanol[73]. To the best of our knowledge, no single iron catalyst has been reported for combined $H_2$ storage and $H_2$ release cycles yet. On the basis of our interest in developing efficient methodologies for $H_2$ storage and utilization by using non-noble metal catalysts, we describe herein a concept of iron promoted partially reversible carbon neutral $H_2$ storage-release cycles in a single device based on dual-functional roles of formamides.

## Results and discussion

### Concept of reversible carbon neutral hydrogen storage-release cycles based on dual-functional roles of formamides

The concept of iron catalyzed reversible carbon neutral hydrogen storage-release cycles based on dual-functional roles of formamides is illustrated in Fig. 2b. Following the hydrogen release pathway (indicated in pink color), formamide ($F_1$) is firstly hydrolyzed into formic acid (FA) and corresponding amine ($A_1$), afterward FA participates in the catalytic cycles of dehydrogenation and hydrogenation. Here the mild potentials of (de)hydrogeantion are provided by redox active iron complexes containing non-innocent pincer ligands[62,64]. It's worth noting that $CO_2$ by-product is captured in situ and stored in the presence of amine ($A_1$) initially liberated from formamide hydrolysis. Even though the individual steps of formamides hydrolysis, FA (or formates) dehydrogenation and their reverse reactions are known, the presented hydrogen storage-release concept enables the reuse of in situ captured

$CO_2$, which allows to (1) retain the hydrogen storage material $CO_2$ in the reaction, therefore, maintain the theoretical hydrogen storage capacity in successive $H_2$ storage-release cycles, (2) avoid undesired carbon release during dehydrogenation processes, and (3) provide superior $H_2$ selectivity/purity compared to other $H_2$ carrier systems. Following the hydrogen storage pathway (indicated in blue color), the stored $CO_2$ can be re-hydrogenated to FA which is then (partially) converted to formamide ($F_1$) via dehydration condensation with corresponding amine ($A_1$). Thanks to the dual-functional roles of formamides, the built-in amine ($A_1$) is beneficial to both $H_2$ storage and $H_2$ release processes by acting as $CO_2$ absorbent, providing a carbon capture and utilization (CCU) strategy to ensure the $H_2$ storage capacity and carbon neutrality of the overall $H_2$ storage-release process.

Following the concept vide supra, both formamide hydrolysis as well as formamide formation were investigated. Thus, initially the hydrolysis process was performed under alkaline conditions[74,75] and a proportional relationship between the base (KOH) loading and FA yields was found (Figs. S1–2). Accordingly, equimolar ratio of base to formamide is necessary to provide a sufficient amount of $H_2$ carrier for the subsequent $H_2$ storage-release cycles. Afterwards, the reaction between different amines and FA to produce formamides was examined (Figs. S3–4)[27]. Interestingly, in this latter condensation process piperazine ($A_3$) gave a much better yield of the corresponding formamides (22%) compared to morpholine ($A_1$, 1%) and piperidine ($A_2$, 1%) under typical reaction conditions used for catalysis (90 °C, 12 h). Obviously, using longer reaction time (72 h) and higher temperature (140 °C) allows to increase the amount of formamide products ($A_1$ 47%, $A_2$ 14%, $A_3$ 46%). Overall, the hydrolysis of formamides to FA and amines is more favored under alkaline condition, than its reverse dehydration condensation.

Next, the $CO_2$ capture effect of those amines was also investigated (Figs. S5-7). Under $CO_2$ pressure (2 bar, 30 min.), both bicarbonate and carbamate species of the corresponding amines were obtained as products in the following order: piperidine ($A_2$, 69%), piperazine ($A_3$, 55%), and morpholine ($A_1$, 42%). These results can be well explained by the reported pKa values of the three amines: $A_2$ (11.22) > $A_3$ (9.73) > $A_1$ (8.36)[76,77]. Under direct air capture conditions (air flow 1.8 L min$^{-1}$, ca. 400 ppm $CO_2$, 36 h), piperazine ($A_3$) led to the highest yield of the corresponding carbamate species (32%) compared to piperidine ($A_2$, 15%) and morpholine ($A_1$, 8%). This is attributed to the stronger hydrogen bonding in piperazine ($A_3$) compared to the other two amines[78]. After all, these results demonstrate the good carbon capture ability of amines $A_1$, $A_2$, and $A_3$, especially with $CO_2$ concentration at ppm level.

## Catalytic hydrogen production based on formamides

Representative non-noble metal pincer complexes (Fig. 3a) were utilized as catalysts in hydrogen production process starting from formamides and the results are summarized in Fig. 3b. Iron pincer-complexes **Fe-1** and **Fe-2**, which were used in formic acid dehydrogenation[64], led to the best yields of $H_2$ 99% and 89% (Figs. S8–9), respectively. Other tested catalysts based on Mn, Co and Mo gave significantly lower $H_2$ yields (up to 43%). In the absence of external base, no $H_2$ was produced. Drastically decreased $H_2$ yields (29% and 37%) were observed after changing the base from KOH to amino acids lysine (Lys) and arginine (Arg), which were recently disclosed for reversible $H_2$ storage-release involving $CO_2$ hydrogenation[17]. Indeed, utilizing stoichiometric amounts of Lys or Arg gave only trace amount of FA due to slower formamide ($F_1$) hydrolysis (1–2% yields, Fig. S2). For hydrogen production also the nature of formamides was examined. Notably, inexpensive and industrially available simple formamides i.e., methanamide (MA) and dimethylformamide (DMF) gave also good $H_2$ yields (78% and 80%, respectively). However, due to practical considerations, e.g. ammonia and dimethylamine are highly

**Fig. 2 | Concept of reversible carbon neutral hydrogen storage-release cycles based on dual-functional roles of formamides. a** Hydrogen content of formic acid and its derivatives (wt%, indicated in green color). **b** Schematic illustration of the concept of pincer-type iron complex catalyzed reversible carbon neutral $H_2$ storage and release based on dual-functional roles of formamides.

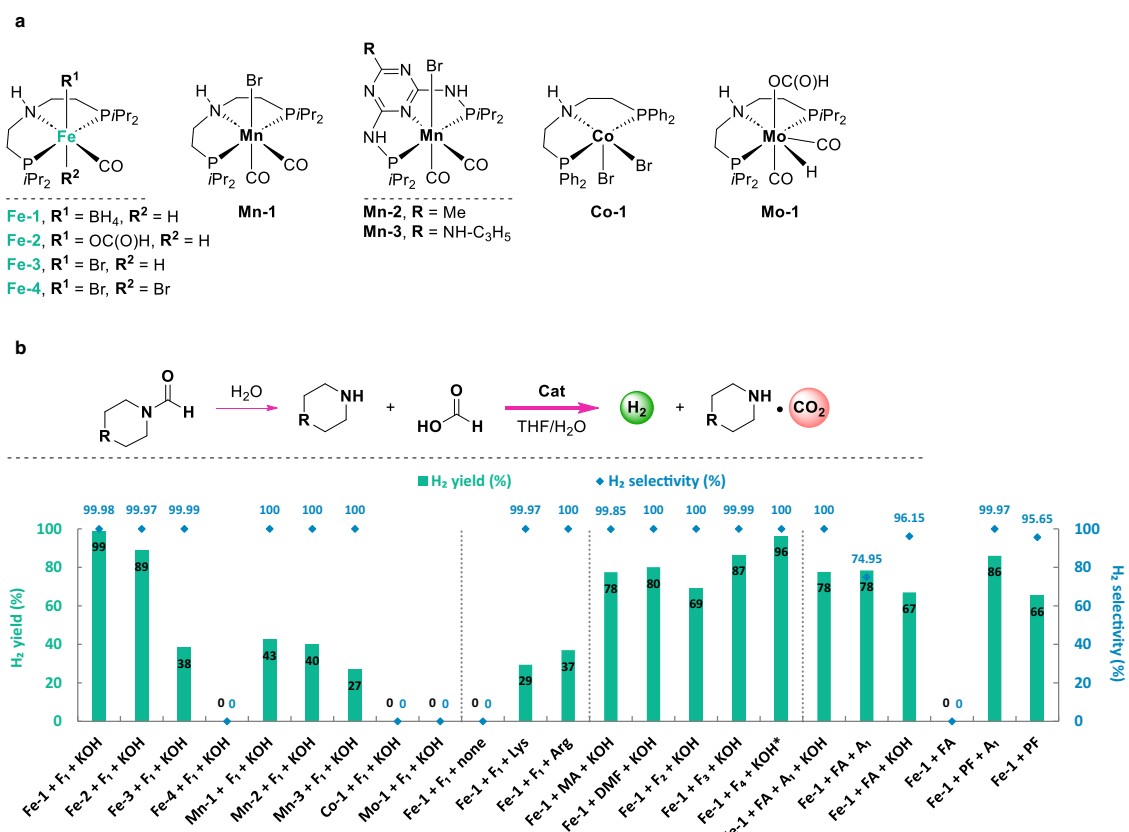

**Fig. 3 | Catalytic hydrogen production from formamides. a** Non-noble metal-based pincer complexes utilized in this study. **b** Comparison of activity under various conditions towards catalytic hydrogen production. Standard conditions: *N*-formylmorpholine ($F_1$, 10 mmol), KOH (10 mmol), catalyst (5 µmol, 500 pm), THF/ $H_2O$ (5/5 mL), 90 °C, 16 h. *1,4-Diformylpiperazine ($F_4$, 5.0 mmol) was used. Yields are based on formyl group in formamides. The dotted lines serve as guides to the eye.

volatile and difficult to handle, we utilized their heavier congeners. As the best candidates, *N*-formylmorpholine ($F_1$) and 1,4-diformylpiper-azine ($F_4$) led to quantitative yields of $H_2$, while *N*-formylpiperidine ($F_2$) and 1-formylpiperazine ($F_3$) gave 69% and 87% $H_2$ yields, respectively.

The base (KOH) loading in catalytic dehydrogenation process was then investigated: in the presence of 25, 50, 75 mol% of KOH, partial $H_2$ yields (37-61%) and lower $H_2$ selectivity were observed (Fig. S10). In the absence of KOH, no conversion of formamide ($F_1$) occurred as indicated by NMR monitoring on the reaction mixture (Fig. S11). Lewis acids are known to assist dehydrogenation processes catalyzed by iron pincer catalysts[64]. However, inferior $H_2$ yield (85%) and selectivity (92.5%) were observed in the presence of 10 mol% $LiBF_4$ compared to the standard conditions (Figs. S10 and S34). Changing THF to other organic solvents, i.e., 2-methyl THF (2-MTHF), ethanol, triglyme, 1,4-dioxane, and DMSO, $H_2$ were observed in 47–74% yields. Using water as sole solvent or under neat conditions, no hydrogen was found due to the low solubility of the catalyst. Decreased $H_2$ yield (87%) was observed by lowering the reaction temperature to 80 °C, while elevated temperature (100 °C) did not promote the reaction but resulted in increased CO concentration (14 ppm; Fig. S41). In all other cases using **Fe-1** complex and formamides, CO was not detected by gas chromatography (below the CO quantification limit of 10 ppm).

### Comparison on different hydrogen carrier systems in catalytic hydrogen production

Under the optimal conditions, the here presented system utilizing formamides is superior regarding both the $H_2$ productivity and selectivity compared to other $H_2$ carriers i.e., formic acid (FA) and potassium formate (PF; Fig. 3b). Specifically, replacing formamide $F_1$ with FA

and amine $A_1$, decreased $H_2$ yields (78%) were observed with $H_2$ selectivity of 100% (in the presence of KOH) and 74.9% (in the absence of KOH). Notably, 80 ppm CO were detected in the $H_2$ storage system of FA and $A_1$ (Fig. S27). Loading FA with KOH, further decreased $H_2$ yield (67%) was observed, while in the presence of FA only, no dehydrogenation occurred. On the other hand, starting from potassium formate (PF), $H_2$ was obtained in yields of 86% (in the presence of $A_1$) and 66% (in the absence of $A_1$).

### Catalytic hydrogen storage in formates and formamides

Next, the process of $H_2$ storage in formates and formamides was investigated by using hydrogenation of $CO_2$ or potassium bicarbonate in the presence of amines as model reactions (Fig. 4a)[27,62,79]. In general, the hydrogenation of $CO_2$ or potassium bicarbonate in the presence of morpholine ($A_1$), piperidine ($A_2$), and piperazine ($A_3$) gave good total yields of formates and formamides (82–100%). Specifically, morpholine ($A_1$) and piperidine ($A_2$) led to comparable results regarding the yields of formates (90-97%) and formamides (2–6%), while piperazine ($A_3$) resulted in a significantly higher amount of formamide product (31%) using $CO_2$ as carbon source (Fig. 4b, left side, Figs. S50-52). It's worth noting that the amine promoted $CO_2$ capture product carbamate was formed as minor species in bicarbonate hydrogenation reaction (Fig. S51), thereby avoiding the release of free carbon dioxide even under basic conditions.

Afterwards, variation of reaction parameters in the hydrogenation step using bicarbonate was performed in the presence of morpholine ($A_1$, Fig. 4b, right side, Fig. S53). Reducing the $H_2$ pressure from 60 bar stepwise to 40, 20, and 10 bar, total yields of formates and formamides decreased from 99 to 43%. Moreover, lowering the reaction

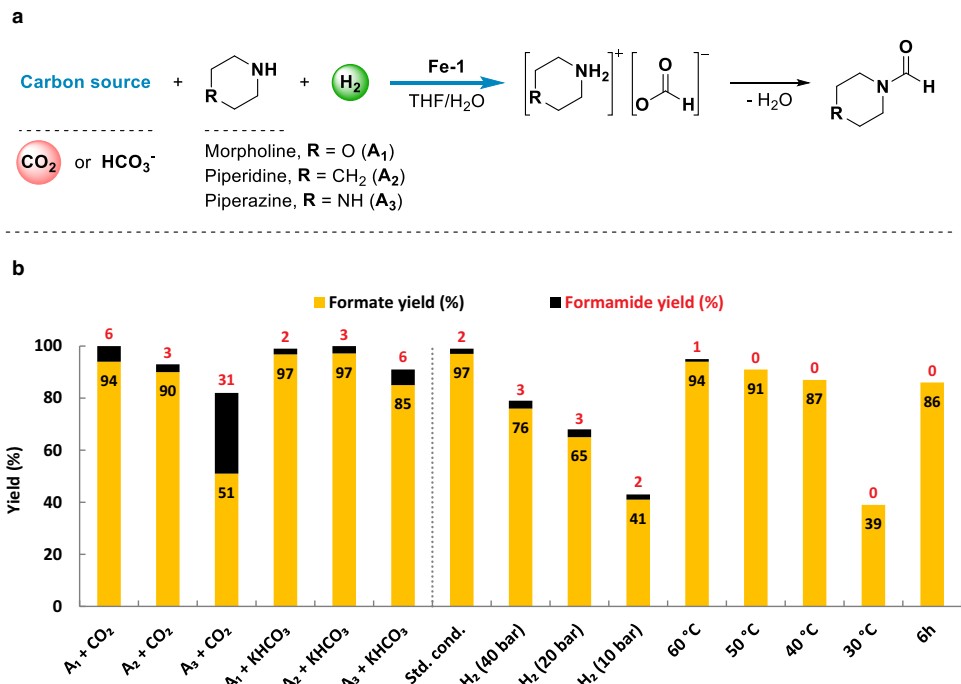

**Fig. 4 | Catalytic hydrogen storage in formates and formamides. a** Catalytic hydrogen storage process via hydrogenation of $CO_2$ or potassium bicarbonate in the presence of amines. **b** Left side: comparison of activity towards hydrogen storage with different amines ($A_1$, $A_2$, and $A_3$) and carbon sources ($CO_2$ and $KHCO_3$). Standard conditions: amine (10 mmol), $CO_2$ (20 bar) or $KHCO_3$ (10 mmol), **Fe-1** (5 μmol, 500 ppm), $H_2$ (60 bar), THF/$H_2O$ (5/5 mL), 90 °C, 12 h. Right side: variation of reaction parameters in hydrogenation of $KHCO_3$ with morpholine. Standard conditions: morpholine ($A_1$, 10 mmol), $KHCO_3$ (10 mmol), **Fe-1** (5 μmol, 500 ppm), $H_2$ (60 bar), THF/$H_2O$ (5/5 mL), 90 °C, 12 h. Yields are based on amine. The dotted lines serve as guides to the eye.

temperature from 90 °C to 60, 50, and 40 °C, no obvious loss of hydrogen storage capacity was observed, while further decrease to 30 °C, drastically dropped the formate yield to 39%. Further, time dependent product generation of hydrogen storage and release reactions catalyzed by **Fe-1** was investigated (Table S1). Lower total yields of formates and formamides were obtained in 3 and 6 h (66% and 87%, respectively) in hydrogenation reactions with morpholine ($A_1$) and $CO_2$. On the other hand, performing the dehydrogenation reactions with *N*-formylmorpholine ($F_1$) in shorter reaction times led to decreased $H_2$ yields (29% in 4 h and 49% in 8 h). These results demonstrate that long reaction times are indeed required.

### Promoting effect of amines in hydrogen storage and release processes

Next, we explored the promoting effect of seven additional amines in formate dehydrogenation and bicarbonate hydrogenation in more detail (Fig. 5). In addition to $A_1$, $A_2$, and $A_3$, classical amines which are widely utilized in $CO_2$ hydrogenation and corresponding dehydrogenation processes were tested (Fig. 5a). In hydrogen production reactions (Fig. 5b), the presence of amines $A_1$, $A_2$, and $A_3$ gave high $H_2$ yields (up to 92%) and selectivity (up to 100%) compared to the one without amine (56% yield and 95% selectivity). Trials with other amines i.e., diazabicycloundecene (DBU)[52,80,81], diazabicyclooctane (DABCO), trihexylamine (THA)[81], and dimethyloctylamine (DMOA)[82,83] resulted in moderate $H_2$ yields (55% to 76%). However, no $H_2$ was produced by using tetramethylguanidine (TMG). Interestingly, the two basic amino acids Lys and Arg led to $H_2$ in 87% and 90% yields, respectively[17,20,84].

In the corresponding hydrogen storage process (Fig. 5c, Fig. S54), amines $A_1$, $A_2$, and $A_3$ gave quantitative yields of formates and formamides[27], while 64% of formate were obtained in the absence of amine. Moreover, DBU, DABCO[85], THA, DMOA, Lys, and Arg led to either lower formate yields (23% to 87%) or even inhibited amide formation. On the other hand, TMG gave nearly quantitative yields of formate and formamide even though it was not active in the $H_2$

production process at all[85]. As there is no obvious direct correlation of pKa of the applied amine and the storage capacity there will be other factors that potentially influence the system, i.e., solubility and boiling point of amines, hydrogen bonding, steric hindrance, catalyst poisoning etc. After considering the $H_2$ productivity and selectivity in dehydrogenation (Fig. 5b) and total yields of formates and formamides in hydrogenation (Fig. 5c), we concluded that morpholine ($A_1$) and piperazine ($A_3$) are the most suitable amine promoters among all other tested amines. Although formate generation dominates at milder conditions (90 °C, 12 h), formamide yields could be improved at higher temperature and longer reaction time (140 °C, 72 h; Fig. S3), therewith formally clothing the formamide-based hydrogen storage cycle. However, due to practicability milder conditions were employed in subsequent catalytic (de)hydrogenation reactions, as this also allows for efficient and partially reversible $H_2$ storage (Fig. 5b, c).

### Carbon neutral hydrogen storage-release cycles based on dual-functional roles of formamides

After having optimized conditions in hand for both elementary steps, (a) $H_2$ release from formamides and (b) corresponding $H_2$ storage process, we turned our attention to the combination of these hydrogenation and dehydrogenation processes in a single device. The overall "carbon neutral" hydrogen cycle was performed in a closed autoclave starting by dehydrogenation of commercially available formamides using the well-designed catalyst **Fe-1** (500 ppm) in the presence of KOH in aqueous THF solution (90 °C, 16 h). Afterwards, the reactor was cooled to room temperature (r.t., 25 °C) and the generated hydrogen was released carefully to the manual burettes and analyzed by GC. Then, the reactor was charged with $H_2$ (60 bar) and heated to 90 °C without changing the reaction mixture ($H_2$ storage step). After the hydrogen uptake stopped (12 h), the overpressure of $H_2$ was released at r.t. and the autoclave was subjected once more to the $H_2$ release step (90 °C, 16 h). Following this procedure, 10 $H_2$ storage-release cycles were performed over 20 days (Fig. 6, Figs. S55−63). Notably, during the

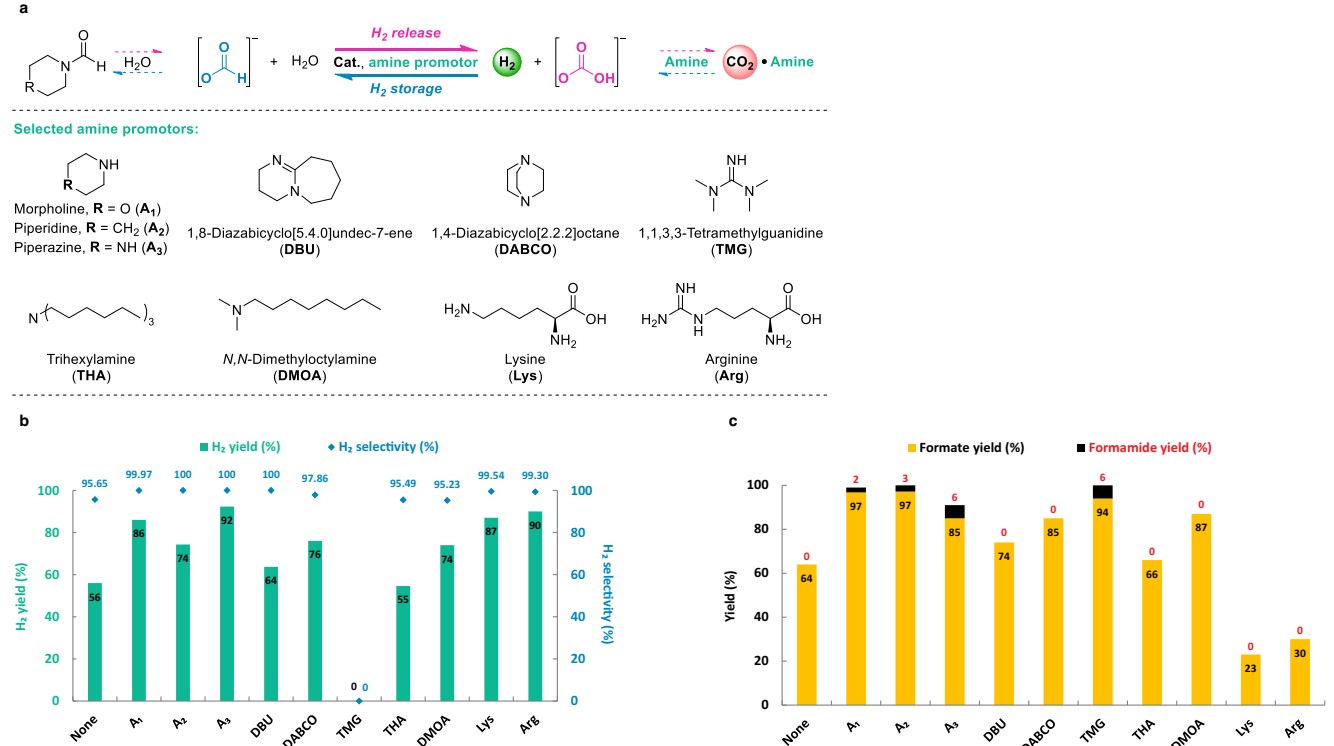

**Fig. 5 | Comparison between selected amine promotors in hydrogen storage and release reactions. a** Chemical structures of selected amine promotors utilized in formate dehydrogenation and bicarbonate hydrogenation. **b** Hydrogen production from formate in the presence of various amines. Standard conditions: $KHCO_2$ (10 mmol), amine (10 mmol), **Fe-1** (5 μmol), $THF/H_2O$ (5/5 mL), 90 °C, 16 h.

Yields are based on $KHCO_2$. **c** Hydrogenation of bicarbonate in the presence of various amines. Standard conditions: $KHCO_3$ (10 mmol), amine (10 mmol), **Fe-1** (5 μmol, 500 ppm), $THF/H_2O$ (5/5 mL), $H_2$ (60 bar), 90 °C, 12 h. Yields are based on $KHCO_3$.

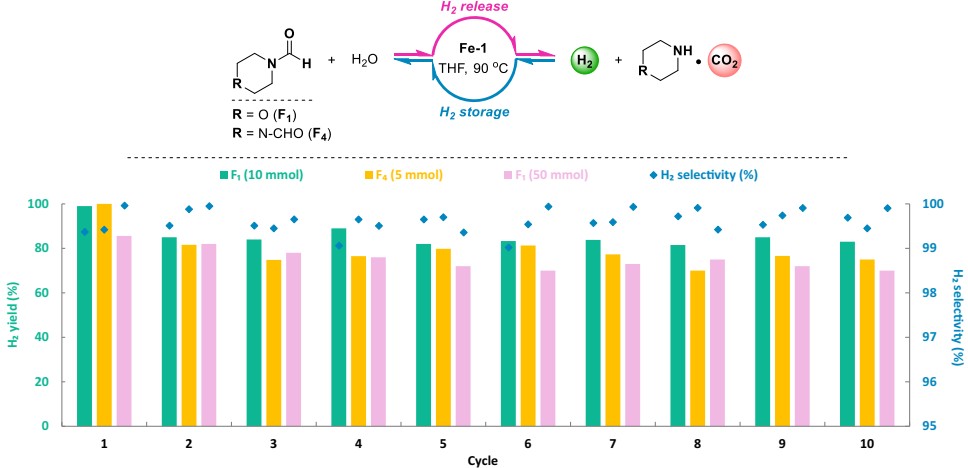

**Fig. 6 | Fe promoted partially reversible carbon neutral hydrogen storage-release cycles using formamides.** Hydrogen evolution in the storage-release cycles applying formamides. Standard conditions: formamide, KOH (1.0 equiv.),

**Fe-1** (500 ppm), $THF/H_2O$ (5/5 mL), 90 °C. The cycles started from dehydrogenation (16 h), then hydrogenation (12 h, 60 bar of $H_2$) was performed. Yields are based on formyl group in formamides.

whole time, only $H_2$ is charged and discharged and the reloading of hydrogen storage material, catalyst, solvents, additives is not necessary. Even though the iron pincer complexes are generally sensitive to air (oxygen), once the $H_2$ storage-release cycles are in operation, the whole system is closed and generally under over-pressure of $H_2$. On the other hand, air has also to be excluded from the system in order to suppress the hydrogen-air explosions (4.0–75.6%v/v of $H_2$ in air).

$^{31}$P NMR spectra of pre- and post-reaction samples (after 1 cycle) revealed that the original signal of **Fe-1** complex (99.6 ppm) was

shifted to lower field (114.0 ppm) after the catalytic dehydrogenation reaction (Fig. S64). This signal is assigned to iron pincer derivative **I-2** (Fig. 2) and considered as the resting state in (de)hydrogenation reactions[64]. Besides, only minor species were found in the spectra which might either be the stereoisomers (e.g., trans- and cis-configurations) of the iron pincer complexes or their decomposition products[86].

Comparing the different tested formamides, 1,4-diformylpiperazine ($F_4$) resulted in higher $H_2$ selectivity (>99.5%) than *N*-

formylmorpholine ($F_1$, >99.0%) at 10 mmol loading due to the better carbon capture ability of the corresponding amine piperazine ($A_3$) compared to morpholine ($A_1$) especially at low $CO_2$ concentration (Fig. S5). Slightly lower $H_2$ yields were observed with $F_4$ (>70%) compared to $F_1$ (>82%) over 10 $H_2$ charge-discharge cycles, due to the lower hydrogen storage capacity using corresponding amines $A_3$ than $A_1$ (Fig. 4b). To our delight, upscaling reactions applying $N$-formylmorpholine ($F_1$, 50 mmol) reached 86% $H_2$ yield in the first cycle, even though gradually decreased yields were observed at 70% in the 10th cycle. Overall, $H_2$ can be obtained in more than 70% yield and 99% selectivity in 10 charge-discharge cycles (Table S2). For a direct application of the generated hydrogen in PEM fuel cells and to avoid the poisoning of platinum electrodes[87], it is important to note that CO was not detected (below the GC quantification limit of 10 ppm) in the $H_2$ stream. Advantageously, both the hydrogenation and dehydrogenation steps operated at a temperature level of 90 °C, which can be supplied by the waste heat from e.g., PEM fuel cells or hydrogen internal combustion engines[88].

In conclusion, we demonstrate partially reversible hydrogen storage-release cycles utilizing formamides. This class of hydrogen storage materials has been largely overlooked despite their attractive physical and chemical properties (inertness, hydrogen content, toxicity, boiling point, etc.). In the presented system, the inherent components of formamides play a dual-functional roles: (a) the formic acid part enables $H_2$ storage and release and (b) the built-in amines provide a carbon capture and utilization (CCU) strategy allowing for an overall "carbon neutral" energy storage system. By using well-designed iron catalyzed hydrogenation and dehydrogenation steps, selective hydrogen formation (CO below detection limit of GC) under mild conditions and high catalyst productivity as well as stability (>20 days) were achieved. To the best of our knowledge, this is also one of the rare examples that an iron based catalytic system allows multiple $H_2$ storage-release cycles in a single device.

Starting from carbon dioxide or bicarbonate in the presence of selected amines, $H_2$ storage proceeded with quantitative total yields of formamides and formates at comparably low temperature (<100 °C). Among the different tested amines, morpholine ($A_1$) and piperazine ($A_3$) exhibited superior behavior in both $H_2$ storage and $H_2$ release processes. The feasibility of combined hydrogenation and dehydrogenation processes in a single device was demonstrated in 10 $H_2$ charge-discharge cycles catalyzed by an iron complex under mild reaction conditions. Advantageously, the presented system is partially reversible and no reloading of hydrogen storage material, catalyst, solvents, additives is necessary during the whole process.

## Methods

### Calculation of the hydrogen contents (wt%)
The hydrogen contents (wt%) of formic acid, formate salts, and formamides are calculated as follows:

$$wt\%_{formic\ acid} = M_{H2}/(M_{formic\ acid}) \times 100\% \qquad (1)$$

$$wt\%_{formate\ salt} = M_{H2}/(M_{formate\ salt} + M_{H2O}) \times 100\% \qquad (2)$$

$$wt\%_{formamide} = (M_{H2} \times N)/(M_{formamide} + M_{H2O} \times N) \times 100\% \qquad (3)$$

where M is the molecular weight, N is the number of formyl groups per formamide molecule.

### Standard procedure for catalytic dehydrogenation starting from formamides
Under an argon atmosphere, $N$-formylmorpholine ($F_1$, 1 mL, 10 mmol), base (10 mmol), catalyst (5 μmol), THF (5 mL) and $H_2O$ (5 mL) were added to a 100 mL autoclave equipped with a magnetic stir bar. Then, the reaction mixture was heated and stirred in a pre-heated oil bath for 16 h. The reactor was cooled to r.t. (25 °C) and the inside pressure was released carefully to the manual burettes. A 5 mL degassed syringe was used to obtain a gas sample analyzed by gas chromatography (GC, CO quantification limit of 10 ppm). Yield of $H_2$ is calculated as follows:

$$Yield_{H_2} = (mmol\ H_2)/(mmol\ formyl\ group\ in\ formamides) \times 100\%$$
$$(4)$$

### Standard procedure for catalytic hydrogenation of $CO_2$ or bicarbonate
Under an argon atmosphere, amine (10 mmol), $CO_2$ (20 bar) or $KHCO_3$ (1 g, 10 mmol), **Fe-1** (2 mg, 5 μmol), THF (5 mL) and $H_2O$ (5 mL) were added to a 100 mL autoclave equipped with a magnetic stir bar. After pressurizing the reactor with $H_2$ (60 bar), the reaction mixture was heated and stirred on a pre-heated oil bath for 12 h. Then, the reactor was cooled to r.t. (25 °C) and the overpressure was carefully released. A biphasic reaction mixture was obtained containing a transparent organic upper layer and an aqueous yellow lower layer. Addition of deionized water (ca. 3 mL) to the above mentioned biphasic mixture resulted in a homogeneous solution[17]. Imidazole (170 mg, 2.5 mmol) was added as an NMR internal standard (I.S.) to the reaction mixture, which was then analyzed by [1]H NMR with ca. 0.1 mL $D_2O$ to lock the signals. Yields of formate and formamide are calculated as follows:

$$Yield_{formate} = (mmol\ formate)/(mmol\ amine) \times 100\% \qquad (5)$$

$$Yield_{formamide}(mmol\ formamide)/(mmol\ amine) \times 100\% \qquad (6)$$

### Standard procedure for catalytic $H_2$ evolution in the $H_2$ storage-release cycles
The $H_2$ storage-release cycles start from the dehydrogenation ($H_2$ release): **Fe-1** (2 mg, 5 μmol, 500 ppm), $N$-formylmorpholine ($F_1$, 1 mL, 10 mmol), KOH (561 mg, 10 mmol), THF (5 mL) and $H_2O$ (5 mL) were added to a 100 mL autoclave equipped with a magnetic stir bar. The reaction mixture was then heated and stirred in a pre-heated oil bath at 90 °C for 16 h. The reactor was cooled to r.t. (25 °C) and the stored hydrogen was released carefully to the manual burettes then the content of the gas phase was analyzed with a 5 mL degassed syringe by gas chromatography (GC, CO quantification limit of 10 ppm). The autoclave was then filled with 60 bar of $H_2$, heated and stirred on a pre-heated oil bath at 90 °C for 12 h ($H_2$ storage). After the completion of $H_2$ storage, the reactor was cooled to r.t. (25 °C) and the overpressure was carefully released. Then the autoclave was subjected to the $H_2$ release step once again. Following such process, the $H_2$ evolution in the $H_2$ storage-release cycles were implemented over 20 days. Yield of $H_2$ is calculated according to Eq. (4).

## Data availability
All data generated or analyzed during this study are included in the published article and its supplementary information files. Data are also available from the Corresponding Author upon request.

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

## Acknowledgements
We acknowledge financial support from the State of Mecklenburg-Vorpommern and European Union (EFRE; project "h2cycle"), and the Leibniz-Program Cooperative Excellence K308/2020 (project "SUPREME"). The authors thank the analytical team of LIKAT for their kind support.

## Author contributions
Conceptualization, D.W. and X.S.; Methodology, D.W. and X.S.; Investigation, D.W. and X.S.; Resources, H.J. and M.B; Writing, D.W., X.S., H.J., C.D., and M.B.; Funding Acquisition, H.J., C.D., and M.B.; Supervision, D.W., H.J., C.D., and M.B. All authors have read and agreed to the published version of this work.

## Funding

## Competing interests
The authors declare no competing interests.
