## [Peer Review File · Nature Communications]

Carbon neutral hydrogen storage and release cycles based on dual-functional role of formamidesREVIEWER COMMENTS

Reviewer #1 (Remarks to the Author):

It is an interesting concept and a very fundamental research. Data is concise and well analyzed.

The pitfall of the approach is in that the reactions are carried out in a mixture of very diluted organic solvent/H₂O and in inert atmosphere of argon. Reported H₂ purities (in high 90s%) aren't unusual if one works with millimolar concentrations

Additionally, the fact that this is the first non-precious metal based system is undercut by the fact that the catalysts have to be stored in the dark and in the inert atmosphere. Preparation of extra dry and pure catalysts is more cost intensive than industrially-relevant precious metal catalysts. As such, applicability of this research would be limited to strictly research-based processes.

CCU is an important and well known technology that is not constrained to organometallic community and one group, and as such, if the comparisons and callouts to this technology are to be made, authors need to make connections to currently utilized technologies and recent breakthroughs (other than academic literature, IEA website may be a good start for this topic).

Nonetheless, the concept presented here and ingenuity of finding another H₂ carrier is an important discovery and in the future may lead to development of a more robust process.

Reviewer #2 (Remarks to the Author):

In this manuscript Junge, Du, Beller and co-workers have reported the use of aqueous formamide as a hydrogen storage system. The high activity of iron pincer catalyst, release of highly pure H₂ and the demonstration of 10 charge-discharge cycles are attractive features of this article. However, I am not convinced that the paper has novelty of the standard needed for the Nat Comm. The dehydrogenation process takes place via two steps: (i) hydrolysis of formamide under alkaline condition to form formate and amine and (ii) dehydrogenation of formate to CO₂. Both of these steps have been independently well studied. For example, see Canadian Journal of Chemistry, Volume 80, Number 10, October 2002, pp. 1343-1350(8), and Angewandte Chemie International Edition, 45: 2893-2897. <https://doi.org/10.1002/anie.200600283> for hydrolysis of formamides and J. Am. Chem. Soc. 2014, 136, 29, 10234–10237 for the dehydrogenation of formate using an iron-Macho pincer catalyst. The reverse reaction should be hydrogenation of CO₂ in the presence of amines to make formamides. However, the reverse reaction mainly leads to the formation of formates as shown in Fig 4. In my opinion, this would not be classified as a reversible system as per the concept of LOHC. There are examples of earth-abundant metal catalysts (manganese and iron) for the hydrogenation of CO₂ and amines to formamides, for example see: ACS Catal. 2017, 7, 9, 6347–6351; ACS Catal. 2018, 8, 2, 1338–1345.

Considering these precedences and concern with the reversibility of the process, I do not recommend the acceptance of this manuscript in Nat. Commun.

Reviewer #3 (Remarks to the Author):

Authors report an elaborate description of reversible catalytic hydrogen storage medium which makes use of formamides as intermediates/bases. The results are novel with respect to the process as a whole - authors state that no single catalyst could perform all hydrogenation/dehydrogenation steps reported here in a stand-alone catalytic system and I concur.

Although some might note that catalysts involved in the study are not new, the process as a whole certainly is and I can highlight the extensive screening performed at the initial stage of the work which will help others in the field.

I believe the work is of high scientific and technical quality and importance and suggest addressing the following questions:

- 1) Can authors clarify the origin of the ammine base effects they describe on page 10? It appears that pKa of the bases is not directly linked to the storage capacity of the system which is a little counterintuitive.
- 2) Iron pincer catalysts are well known to benefit from Lewis acid promotion in formate dehydrogenation. Have authors utilized this in the case of formamides?
- 3) I believe that kinetic measurements are needed to demonstrate the catalyst stability implied in Figure 6. If the performance of the catalyst is indeed undegraded throughout the cycles then authors' claim is significantly more solid.
- 4) As with many iron pincers, the stability of the catalyst might be limited. Authors can provide pre- and postreaction ^{31}P NMR to elaborate on the catalyst integrity throughout cycling experiment. This is also important for the peers working on deactivation mechanisms.

REVIEWER COMMENTS

Reviewer #1 (Remarks to the Author):

It is an interesting concept and a very fundamental research. Data is concise and well analyzed.

The pitfall of the approach is in that the reactions are carried out in a mixture of very diluted organic solvent/H₂O and in inert atmosphere of argon. Reported H₂ purities (in high 90s%) aren't unusual if one works with millimolar concentrations

The authors thank the reviewer for the comments. Actually, the catalytic dehydrogenation reaction couldn't take place under neat conditions or with H₂O as sole solvent (see Supplementary Figure S10). Therefore, it's necessary to introduce organic solvent, preferably THF here, to increase the catalytic activity. The corresponding discussion is now revised as following: *"Using water as sole solvent or under neat conditions, no hydrogen was found due to the low solubility of the catalyst."*

The presented 99%+ H₂ purities using formamides as hydrogen storage materials is superior compared to the one using formic acid where equimolar CO₂ is normally released together with H₂. Besides, the hydrogen storage-release cycles could be scaled-up to 50 mmol while keeping the H₂ purities at around 99.9%.

For the comment of "in inert atmosphere of argon", please see the response to the following remark.

Additionally, the fact that this is the first non-precious metal based system is undercut by the fact that the catalysts have to be stored in the dark and in the inert atmosphere. Preparation of extra dry and pure catalysts is more cost intensive than industrially-relevant precious metal catalysts. As such, applicability of this research would be limited to strictly research-based processes.

The authors thank the reviewer for the comments. Indeed, the iron pincer complexes are generally sensitive to oxygen, therefore the catalytic (de)hydrogenation reactions have to be set up in the inert atmosphere. However, once the H₂ storage-release cycles are in operation, the whole system is closed and always under over-pressure of pure H₂. On the other hand, air has also to be excluded from the system in order to suppress the hydrogen-air explosions (4.0 - 75.6%v/v of H₂ in air). Besides, the catalytic reactions utilize H₂O/THF as co-solvent, therefore it's not necessary to prepare the system under extra dry conditions, thus simplifying the hydrogen storage process. Corresponding discussion is now added as following:

"Even though the iron pincer complexes are generally sensitive to air (oxygen), once the H₂ storage-release cycles are in operation, the whole system is closed and generally under over-pressure of H₂. On the other hand, air has also to be excluded from the system in order to suppress the hydrogen-air explosions (4.0% - 75.6%v/v of H₂ in air)."

CCU is an important and well known technology that is not constrained to organometallic community and one group, and as such, if the comparisons and callouts to this technology are to be made, authors need to make connections to currently utilized technologies and recent breakthroughs (other than academic literature, IEA website may be a good start for this topic).

The authors thank the reviewer for the comment. Corresponding discussion about currently applied CCU technologies is now added as following:

"As one of the most prominent examples of CCU, the "George Olah Methanol Plant" in Iceland is based on local renewable energy and CO₂.²⁶ Its total electrical energy demand and the overall efficiency reach 9.5 MWh/t methanol and 60%."

Nonetheless, the concept presented here and ingenuity of finding another H₂ carrier is an important discovery and in the future may lead to development of a more robust process.

Reviewer #2 (Remarks to the Author):

In this manuscript Junge, Du, Beller and co-workers have reported the use of aqueous formamide as a hydrogen storage system. The high activity of iron pincer catalyst, release of highly pure H₂ and the demonstration of 10 charge-discharge cycles are attractive features of this article. However, I am not convinced that the paper has novelty of the standard needed for the Nat Comm. The dehydrogenation process takes place via two steps: (i) hydrolysis of formamide under alkaline condition to form formate and amine and (ii) dehydrogenation of formate to CO₂. Both of these steps have been independently well studied. For example, see Canadian Journal of Chemistry, Volume 80, Number 10, October 2002, pp. 1343-1350(8), and Angewandte Chemie International Edition, 45: 2893-2897. <https://doi.org/10.1002/anie.200600283> for hydrolysis of formamides and J. Am. Chem. Soc. 2014, 136, 29, 10234–10237 for the dehydrogenation of formate using an iron-Macho pincer catalyst.

The authors thank the reviewer for the comments. We agree that the two steps: formamides hydrolysis and formates dehydrogenation are known in literature. However, what we exclusively show in the current study is that starting from formamides highly pure H₂ (>99.9%) can be released during dehydrogenation reactions. This is in contrast to previous literature that reported application of formic acid or formates generally producing mixtures of CO₂ and H₂ in the dehydrogenation processes. Therefore, the presented hydrogen storage method might contribute to make hydrogen a relatively cleaner and more efficient energy carrier. Corresponding discussion is now added as following:

“Even though the individual steps of formamides hydrolysis, FA (or formates) dehydrogenation and their reverse reactions are known, the presented hydrogen storage-release concept enables the reuse of in situ captured CO₂, which allows to 1) retain the hydrogen storage material CO₂ in the reaction, therefore, maintain the theoretical hydrogen storage capacity in successive H₂ storage-release cycles, 2) avoid undesired carbon release during dehydrogenation processes, and 3) provide superior H₂ selectivity/purity compared to other H₂ carrier systems.”

The above-mentioned publications have been cited as references 65, 75, and 76 in the manuscript.

The reverse reaction should be hydrogenation of CO₂ in the presence of amines to make formamides. However, the reverse reaction mainly leads to the formation of formates as shown in Fig 4. In my opinion, this would not be classified as a reversible system as per the concept of LOHC. There are examples of earth-abundant metal catalysts (manganese and iron) for the hydrogenation of CO₂ and amines to formamides, for example see: ACS Catal. 2017, 7, 9, 6347–6351; ACS Catal. 2018, 8, 2, 1338–1345.

The authors thank the reviewer for raising this point and fully agree that the hydrogenation of CO₂ and amines to formamides is known in literature. In manuscript Figure 4, we show that the hydrogenation of CO₂ or bicarbonate in the presence of amines leads to quantitative total yields of formates and formamides, which means that CO₂ and bicarbonate can quantitatively store H₂, preferably in the form of formates than formamides under relatively mild reaction conditions (90 °C, 12 h). On the other hand, we interpreted that the yields of formamides could be improved under harsher conditions (140 °C, 72 h, Supplementary Figure S3). Nevertheless, the milder conditions were adopted in catalytic reactions rather than the harsher ones, so H₂ can be stored and released more efficiently and reversibly in the hydrogenation-dehydrogenation cycles, although the selectivity towards formamides is lower than that of formates. Corresponding discussion is added as following:

“Although formate generation dominates at milder conditions (90 °C, 12 h), formamide yields could be improved at higher temperature and longer reaction time (140 °C, 72 h; Fig. S3), therewith formally clothing the formamide-based hydrogen storage cycle. However, due to practicability milder conditions were employed in subsequent catalytic (de)hydrogenation reactions, as this also allows for efficient and reversible H₂ storage (Figs. 5b-c).”

The above-mentioned publications have been cited as references 27 and 80 in the manuscript.

Considering these precedences and concern with the reversibility of the process, I do not recommend the acceptance of this manuscript in Nat. Commun.

Reviewer #3 (Remarks to the Author):

Authors report an elaborate description of reversible catalytic hydrogen storage medium which makes use of formamides as intermediates/bases. The results are novel with respect to the process as a whole - authors state that no single catalyst could perform all hydrogenation/dehydrogenation steps reported here in a stand-alone catalytic system and I concur.

Although some might note that catalysts involved in the study are not new, the process as a whole certainly is and I can highlight the extensive screening performed at the initial stage of the work which will help others in the field.

I believe the work is of high scientific and technical quality and importance and suggest addressing the following questions:

1) Can authors clarify the origin of the ammine base effects they describe on page 10? It appears that pK_a of the bases is not directly linked to the storage capacity of the system which is a little counterintuitive.

The authors thank the reviewer for raising this question. Unlike the normal acid-base reactions, the catalytic (de)hydrogenation reactions usually involve: substrates coordination, chemical bonds activation/splitting and products dissociation. Therefore, the amine base effects here are not just depending on their pK_a but also other factors/properties like solubility, boiling point, hydrogen bonding, steric hindrance, and poisoning effect to the catalyst and so on. This could not be clarified so far. To make it easier for the audience, corresponding discussion including pK_a values of amines are added in the main text (Figure 5a) as following:

“As there is no obvious direct correlation of pK_a of the applied amine and the storage capacity there will be other factors that potentially influence the system, i.e., solubility and boiling point of amines, hydrogen bonding, steric hindrance, catalyst poisoning etc.”

2) Iron pincer catalysts are well known to benefit from Lewis acid promotion in formate dehydrogenation. Have authors utilized this in the case of formamides?

The authors thank the reviewer for raising this question. We have performed the corresponding experiment using LiBF₄ (10 mol%) as Lewis acid, however inferior H₂ yield (85%) and selectivity (92.5%) were observed compared to the optimized conditions. Corresponding result is now added in the Supplementary Figure S10 and interpreted in the manuscript as following:

“Lewis acids are known to assist dehydrogenation processes catalyzed by iron pincer catalysts.⁶⁵ However, inferior H₂ yield (85%) and selectivity (92.5%) were observed in the presence of 10 mol% LiBF₄ compared to the standard conditions (Figs. S10 and S34).”

3) I believe that kinetic measurements are needed to demonstrate the catalyst stability implied in Figure 6. If the performance of the catalyst is indeed undegraded throughout the cycles then authors' claim is significantly more solid.

The authors thank the reviewer's suggestion. The above-mentioned experiments have been carried out and corresponding results are summarized in Supplementary Table S1 and discussed in the manuscript as following:

"Further, time dependent product generation of hydrogen storage and release reactions catalyzed by Fe-1 was investigated (Table S1). Lower total yields of formates and formamides were obtained in 3 and 6 h (66% and 87%, respectively) in hydrogenation reactions with morpholine (A₁) and CO₂. On the other hand, performing the dehydrogenation reactions with N-formylmorpholine (F₁) in shorter reaction times led to decreased H₂ yields (29% in 4 h and 49% in 8 h). These results demonstrate that long reaction times are indeed required."

4) As with many iron pincers, the stability of the catalyst might be limited. Authors can provide pre- and postreaction ³¹P NMR to elaborate on the catalyst integrity throughout cycling experiment. This is also important for the peers working on deactivation mechanisms.

The authors fully agree with reviewer's comment. The mentioned experiments have been performed. Indeed, a different signal in ³¹P NMR spectra was observed at around 114 ppm after the catalytic dehydrogenation reaction (see Supplementary Figure S64), and is assigned to iron pincer derivative I-2 (see manuscript Figure 2 and J. Am. Chem. Soc. 2014, 136, 10234), which is considered as the resting state in (de)hydrogenation reactions. Corresponding discussion including the stability of iron pincer complexes is added as following:

"³¹P NMR spectra of pre- and post-reaction samples (after 1 cycle) revealed that the original signal of Fe-1 complex (99.6 ppm) was shifted to lower field (114.0 ppm) after the catalytic dehydrogenation reaction (Fig. S64). This signal is assigned to iron pincer derivative I-2 (Fig. 2) and considered as the resting state in (de)hydrogenation reactions.⁶⁵ Besides, only minor species were found in the spectra which might either be the stereoisomers (e.g., trans- and cis-configurations) of the iron pincer complexes or their decomposition products.⁸⁷"

REVIEWERS' COMMENTS

Reviewer #1 (Remarks to the Author):

Reviewers' comments has been satisfactorily addressed. Publish as is.

Reviewer #2 (Remarks to the Author):

The authors have responded to my comments to sufficient extent justifying the significance of their work. The only minor suggestion I have is to use the term "partially reversible" or "possibly reversible" instead of reversible, for example in the abstract (... new system for reversible carbon neutral hydrogen storage...) and where relevant in the manuscript considering that the system is not completely reversible.

Reviewer #3 (Remarks to the Author):

I am fully satisfied with additions to the revised manuscript and believe it is suitable for publication in its current form.

REVIEWERS' COMMENTS

Reviewer #1 (Remarks to the Author):

Reviewers' comments has been satisfactorily addressed. Publish as is.

Reviewer #2 (Remarks to the Author):

The authors have responded to my comments to sufficient extent justifying the significance of their work. The only minor suggestion I have is to use the term "partially reversible" or "possibly reversible" instead of reversible, for example in the abstract (... new system for reversible carbon neutral hydrogen storage...) and where relevant in the manuscript considering that the system is not completely reversible.

The authors thank the reviewer for the suggestion. The term "reversible" has been replaced by "partially reversible" in the abstract and where relevant in the manuscript.

Reviewer #3 (Remarks to the Author):

I am fully satisfied with additions to the revised manuscript and believe it is suitable for publication in its current form.